# Targeting Glucose Metabolism Enzymes in Cancer Treatment: Current and Emerging Strategies

**DOI:** 10.3390/cancers14194568

**Published:** 2022-09-21

**Authors:** Yi Zhang, Qiong Li, Zhao Huang, Bowen Li, Edouard C. Nice, Canhua Huang, Liuya Wei, Bingwen Zou

**Affiliations:** 1Department of Radiation Oncology, Division of Thoracic Oncology, Cancer Center, West China Hospital, Sichuan University, Chengdu 610041, China; 2West China School of Basic Medical Sciences and Forensic Medicine, State Key Laboratory of Biotherapy and Cancer Center, and West China Hospital, Sichuan University, and Collaborative Innovation Center for Biotherapy, Chengdu 610041, China; 3Department of Biochemistry and Molecular Biology, Monash University, Clayton, VIC 3800, Australia; 4School of Pharmacy, Weifang Medical University, Weifang 261053, China

**Keywords:** malignant tumor, glucose metabolism enzymes, glycolysis, targeted therapy

## Abstract

**Simple Summary:**

Reprogramming of glucose metabolism is a hallmark of cancer and can be targeted by therapeutic agents. Some metabolism regulators, such as ivosidenib and enasidenib, have been approved for cancer treatment. Currently, more advanced and effective glucose metabolism enzyme-targeted anticancer drugs have been developed. Furthermore, some natural products have shown efficacy in killing tumor cells by regulating glucose metabolism, offering novel therapeutic opportunities in cancer. However, most of them have failed to be translated into clinical applications due to low selectivity, high toxicity, and side effects. Recent studies suggest that combining glucose metabolism modulators with chemotherapeutic drugs, immunotherapeutic drugs, and other conventional anticancer drugs may be a future direction for cancer treatment.

**Abstract:**

Reprogramming of glucose metabolism provides sufficient energy and raw materials for the proliferation, metastasis, and immune escape of cancer cells, which is enabled by glucose metabolism-related enzymes that are abundantly expressed in a broad range of cancers. Therefore, targeting glucose metabolism enzymes has emerged as a promising strategy for anticancer drug development. Although several glucose metabolism modulators have been approved for cancer treatment in recent years, some limitations exist, such as a short half-life, poor solubility, and numerous adverse effects. With the rapid development of medicinal chemicals, more advanced and effective glucose metabolism enzyme-targeted anticancer drugs have been developed. Additionally, several studies have found that some natural products can suppress cancer progression by regulating glucose metabolism enzymes. In this review, we summarize the mechanisms underlying the reprogramming of glucose metabolism and present enzymes that could serve as therapeutic targets. In addition, we systematically review the existing drugs targeting glucose metabolism enzymes, including small-molecule modulators and natural products. Finally, the opportunities and challenges for glucose metabolism enzyme-targeted anticancer drugs are also discussed. In conclusion, combining glucose metabolism modulators with conventional anticancer drugs may be a promising cancer treatment strategy.

## 1. Introduction

The primary physiological function of glucose is to serve as a source of carbon and energy for the body’s important activities for meeting the needs of cell growth and proliferation. There are three main glucose energy conversion pathways: aerobic oxidation, anaerobic oxidation (glycolysis), and the pentose phosphate pathway (PPP). Glycolysis is defined as the breakdown of glucose or glycogen into lactate accompanied by the production of small amounts of adenosine triphosphate (ATP) under hypoxic conditions. Oxidative phosphorylation (OXPHOS) and anaerobic glycolysis are the two major catabolic glucose pathways, in which glycolysis is the common initiation pathway of both [1]. Glucose enters the cell through glucose transferase (GLUT) and produces pyruvate by the functions of three rate-limiting enzymes, hexokinase (HK), phosphofructokinase (PFK), pyruvate kinase (PK), as well as other non-rate-limiting enzymes. Under normal oxygen concentrations, pyruvate enters the mitochondria for oxidative decarboxylation to produce acetyl coenzyme A, followed by complete oxidation by a series of rate-limiting and non-rate-limiting enzymes to produce energy. Briefly, glucose is completely broken down through glycolysis, the tricarboxylic acid cycle (TCA cycle), and OXPHOS (Figure 1A). In the presence of oxygen, one molecule of glucose can yield a net production of 36 to 38 molecules of ATP if processed via this three-stage pathway, the most critical pathway in cellular metabolism. Under anaerobic conditions, pyruvate produced by normal cells through the glycolysis pathway will no longer enter the TCA cycle. However, it produces lactate in the cytoplasm through lactate dehydrogenase (LDH), which produces less ATP.

In the 1920s, Otto Warburg first observed that cancer cells tend to metabolize glucose to lactate even in the presence of sufficient oxygen, known as the Warburg effect or aerobic glycolysis (Figure 1B) [2]. By the 1980s, with the application of fluorodeoxyglucose positron emission tomography (FDG-PET), glucose uptake in clinical tissue samples could be imaged, and the Warburg effect was confirmed in almost all cancers [3,4,5]. The Warburg effect promotes the glucose uptake of cancer cells in the tumor microenvironment [6]. Further studies have revealed that the tumor growth rate positively correlates with glucose levels and that high glucose levels in cancer patients are associated with poor prognosis [7,8,9]. Thus, cancer starvation therapy based on glucose deprivation is emerging as an effective treatment for suppressing tumor growth [10,11,12]. For example, the ketogenic diet can inhibit the metabolic proliferation of cancer cells by reducing blood glucose [13,14,15]. The Warburg effect is mainly a compensatory activity of cancer cells to adapt to the tumor microenvironment (TME). On one hand, high-efficiency aerobic glycolysis contributes to the proliferation of cancer cells by allowing cancer cells to produce abundant ATP. Although the energy produced by each glucose molecule during aerobic glycolysis is less than that produced by OXPHOS, aerobic glycolysis can generate a number of ATP molecules comparable to OXPHOS when the amount of glucose is sufficient [16]. On the other hand, aerobic glycolysis provides cells with intermediates required for biosynthetic pathways, including ribose for nucleotide synthesis and glycerol, citrate, and nonessential amino acids for lipid synthesis. For example, glucose-6-phosphate (G-6-P) is a substrate for the pentose phosphate pathway that produces reduced nicotinamide adenine dinucleotide phosphate (NADPH) and ribose-5-phosphate (R-5-P) substrates, and R-5-P is a substrate for nucleic acid synthesis. Additionally, 3-phosphoglyceric acid is the main precursor substance for serine and glycine synthesis, and serine is involved in one-carbon unit metabolism and is closely related to the production of purines, thymidine, and NADPH [17], which protects cancer cells from damage induced by oxidative stress [2]. Therefore, the Warburg effect is beneficial to the bioenergetics and biosynthesis of cancer cells. In addition, aerobic glycolysis also brings other benefits to cancer cells. For example, a large amount of pyruvate is converted into lactate without entering the TCA cycle to complete OXPHOS during aerobic glycolysis in cancer cells. On one hand, this can reduce the production of reactive oxygen species (ROS) and thus protect mitochondria [2]. On the other hand, long-term maintenance of moderate levels of ROS boosts cancer progression [2]. Meanwhile, glycolysis eventually creates a high-lactate, low-glucose TME. Immunosuppressive cells such as myeloid-derived suppressor cells (MDSCs) [18] and regulatory T (Treg) cells present better tolerance in high-lactate, low-glucose environments. Treg cells can maintain their function by oxidizing lactate [19]. Lactate participates in the homeostatic regulation of M1 macrophages [20] and inhibits CD8^+^ T cells and natural killer cells (NK cells) from producing γ-interferon [21], which in turn maintains an immunosuppressive microenvironment [22,23]. Overall, metabolic reprogramming of glucose metabolism, namely the Warburg effect, provides cancer cells with the energy, substrates, and environment required for their survival and contributes significantly to cancer development.

Extensive studies have confirmed that metabolic reprogramming of glucose, which plays a vital role in the proliferation, invasion and metastasis of cancer cells, is closely associated with the survival of cancer cells [24,25]. Therefore, metabolic reprogramming of glucose metabolism is considered the essential hallmark of tumorigenesis and development [26]. Further research found that there are a large number of therapeutic targets for aerobic glycolysis, mainly including key enzymes and transporters. Thus, targeting aerobic glycolysis in cancer cells is a promising therapeutic strategy. Numerous studies have found that targeted intervention in the aerobic glycolysis of cancer cells can inhibit cancer growth. Several aerobic glycolysis inhibitors are under investigation in preclinical and clinical studies. A few of them, such as ivosidenib and enasidenib, have been successfully translated into clinical applications for cancer treatment [27]. However, toxicity and inferior anticancer efficacy still hinder clinical translation. With the rapid development of chemical technology, more advanced and effective glucose metabolism enzyme-targeted anticancer drugs have been developed. Additionally, several studies found that some natural products could suppress cancer progression by regulating glucose metabolism enzymes. Overall, significant progress has been made in recent years in developing anticancer therapeutics targeting metabolic enzymes.

In this review, we briefly introduce normal and reprogrammed glucose metabolism in cancer cells. Furthermore, we focus on enzymes that can serve as therapeutic targets, which may help to develop new anticancer strategies. In addition, this review will present the latest studies on emerging candidate agents targeting glucose metabolism enzymes that could be used in cancer treatment, including small-molecule inhibitors and natural products. Finally, the opportunities and challenges for glucose metabolism enzyme-targeting anticancer drugs are also discussed. This paper aims to highlight the importance of glucose metabolism regulators as valuable tools for developing new anticancer therapies.

## 2. Drugs That Target Glucose Metabolism Enzymes

To meet the demand for reagents and energy for the rapid and continuous cell proliferation in tumor development and progression, multiple metabolic pathways are changed in tumor cells to promote proliferation, among which abnormal glucose metabolism is the most classic and prominent feature. Therefore, inhibition of abnormal glucose metabolism can inhibit cancer growth. Glucose metabolism enzymes as therapeutic targets may provide a novel perspective and insight for cancer treatment. In the last decade, with the rapid development of medicinal chemistry, several glucose metabolism enzyme inhibitors have been, and continue to be, developed as anticancer drugs. In this section, we review the glucose metabolism enzymes that could serve as therapeutic targets, as shown in Figure 2. Meanwhile, we systematically summarize the current and emerging drugs targeting glucose metabolism enzymes, which may provide fresh ideas for developing anticancer drugs.

### 2.1. Drugs Targeting Glucose Transferase (GLUT)

Cancer cells consume large amounts of glucose for glycolysis, and glucose enters the cytoplasm through the phospholipid bilayer with the help of GLUT [28]. The GLUT family has 14 members, all of which are capable of selectively transporting different sugar molecules [28]. Among them, GLUT1, GLUT2 (SLC2A2), GLUT3 (SLC2A3), and GLUT4 (SLC2A4) are the four most well-known subtypes, which have distinct regulatory mechanisms and kinetic characteristic and each subtype plays a specific function in maintaining cellular and organismal glucose homeostasis [29,30]. GLUT1 is a widely distributed glucose transporter whose expression is regulated by hypoxia-inducible factor-1α (HIF-1α) [31,32]. GLUT1 has a high affinity for glucose and is highly expressed in a variety of cancers, including lung cancer, prostate cancer, kidney cancer, and lymphoma [33]. In most cancers, a hypoxic TME induces high expression of GLUT1, which enhances the glucose uptake of cancer cells [34]. In addition, high GLUT2 and GLUT3 expression is also simultaneously found in cancer cells [35,36,37]. Multiple myeloma mainly expresses GLUT4, which is responsible for maintaining adequate glucose uptake [33,38]. The uptake of hexoses, such as fructose and mannose, is also significantly increased in cancers as a result of rapid glucose depletion, with GLUT5 specifically transporting fructose in lymphomas and mannose sharing a transport enzyme with glucose [39,40].

Cytochalasin B (Figure 3A), a cell-permeable mycotoxin, was the first molecule identified to inhibit GLUT1, which reduces glucose uptake in hepatocellular carcinoma cells [41]. Since then, a series of GLUT inhibitors have been discovered, including synthetic small-molecule inhibitors and natural products. The small-molecule compounds that inhibit GLUT include STF-31, WZB117, BAY-876, and CG-5. Through high-throughput screening, Chan et al. [42] first found that STF-31 (Figure 3B) could inhibit the growth of renal cancer cells by directly binding to GLUT1 to inhibit glucose uptake. However, normal cells do not rely on glycolysis to provide energy and can take up glucose through other isoforms such as GLUT2. Therefore, STF-31 is non-toxic to normal tissues and can selectively kill cancer cells [43,44]. WZB117 (Figure 3C) is a bishydroxybenzoate compound that inhibits the growth of cancer cells by blocking glucose transport through binding to the glucose binding site of GLUT1 [45,46,47,48]. Moreover, WZB117 can be used in combination with other anti-cancer drugs, such as paclitaxel or cisplatin, to produce synergistic effects on lung and breast cancer cells [46]. Siebeneicher H et al. [49] screened BAY-876 (Figure 3D) from a compound library using high-throughput screening. BAY-876 inhibited GLUT1 with good metabolic stability in vitro, had a high oral bioavailability in vivo, and its anticancer activity was demonstrated in a variety of cancers, including ovarian and triple-negative breast cancer [50,51]. CG-5 is a thiazolidinedione derivative that inhibits GLUT, blocks glucose transport in T cells, and inhibits glycolysis, thus inhibiting the differentiation of Th1 and Th17 cells, inducing differentiated Treg cells, and suppressing the proliferation of CD4^+^ T cells [52]. Although the anticancer effects of GLUT inhibitors such as WZB117, BAY876, and CG-5 have been demonstrated in several tumor models, studies on the safety and side effects of these inhibitors are still limited [43,49,53]. With the advancement of technology, novel GLUT inhibitors, such as PUG-1 (Figure 3E) [54], chromopynones (Figure 3F) [55,56,57], rapaglutin A [35,58], EF24 [59], ketoximes [36], polyphenolic esters [60], pyrazolo-pyrimidines [37], quinazolines [61], phenylalanine amides [62] and many more [35,58,63,64,65,66,67], EF24 [59], ketoximes [36], polyphenolic esters [60], pyrazolo-pyrimidines [37], quinazolines [61], phenylalanine amides [62], and many more [63,64,65,66,67], have emerged but need to be more deeply investigated.

Several natural products have also been shown to suppress growth in cancer cells by inhibiting GLUT, including phloretin, genistein, fasentin, and apigenin. The polyphenol phloretin (Figure 3G) was shown to inhibit GLUT2 in triple-negative breast cancer, leading to the suppression of cancer growth and metastasis [53]. Additionally, phloretin can inhibit GLUT1, which is overexpressed in the hypoxic area of resistant colon cancer cell lines, and induce apoptosis by activating p53-mediated signaling, leading to suppression of growth in resistant cancer cells [68]. Ji et al. [69] demonstrated that genistein (Figure 3H) could induce apoptosis and inhibit the proliferation of renal cancer cells by increasing CDKN2a expression levels and decreasing methylation, suggesting that genistein is also a potential therapeutic agent for cancer. Moreover, as a GLUT inhibitor, genistein can regulate miR-1260b to affect the Wnt signaling pathway to inhibit cancer tissue growth and metastasis [70]. Fasentin (Figure 3I) and its analogs have been shown to inhibit glucose uptake and decrease resistance to caspase activation, which is involved in the chemoresistance of cancer cells [71,72,73]. In addition, fasentin resists angiogenesis via glucose-independent metabolism [72]. For most malignant tumors, angiogenesis, a hallmark of cancer, is not only a significant feature but also a drug target [74]. The dual mechanism of fasentin may change the current state of the treatment for malignant tumors [72]. Inhibition of proliferation and apoptosis induction of cancer cells by apigenin (Figure 3J) were associated with the downregulation of GLUT1 expression, which was partly dependent on the inhibition of HIF-1α [31]. Furthermore, apigenin reduced VEGF secretion by cancer cells under both normoxia and hypoxia, suggesting its potential to inhibit cancer metastasis [75,76,77]. In addition, natural products such as trehalose (Figure 3K) [78], silibinin (Figure 3L) [79], curcumin (Figure 3M) [80], resveratrol (Figure 3N) [81], naringenin [82], quercetin [83], isoquercetin [84], kaempferol [85], xanthohumol [86], caffeine [87], bezielle [88], theophylline [89], (+)-Cryptocaryone [90], and melatonin [91] also have inhibitory effects on GLUT1. Moreover, natural product compounds generally have better safety and less toxicity in comparison to synthetic drugs. For example, Baeckea frutescens leaf extracts could inhibit tumor growth by reducing glucose uptake in breast cancer cells, but there was no obvious cytotoxic effect on normal cells [92]. However, various natural products usually have the limitations of low stability and solubility in the physiological environment and low delivery efficiency due to multi-targeting and low site-specific distribution in the lesion. Thus, drug delivery systems have been designed to improve those disadvantages, such as liposomes, inorganic metal frames, and hydrogels [93,94,95]. Computational modeling and computer-aided drug design have contributed immensely to the successful development of drugs, especially in the contemporary pharmaceutical and drug industries. Integrating computer-aided drug design (CADD) into the development of GLUT has contributed to the enhancement of selective drug targeting with reduced toxicity and off-target effects [96,97,98]. However, few studies exist directly comparing the efficacy and safety of synthetic chemicals against natural substances for targeting glucose transferase (GLUT). We expect that these issues will attract more attention and contribute to intensive research.

In conclusion, GLUT inhibitors demonstrate the potential value of glucose transferase as targets for cancer therapy. Exploring their mechanisms can help to better understand the process of cancer development and progression and develop corresponding targeted drugs. However, the widespread expression of GLUTs in normal cells limits the application of such drugs. Therefore, developing highly selective inhibitors of GLUT to avoid the side effects caused by inhibition of other isoforms is a central challenge in the development of this drug. In addition, the combination of GLUT inhibitors with GLUT signaling pathway inhibitors (e.g., Akt, mTOR, PI3K, HIF-1α, and AMPK) could be a new direction for cancer therapy.

### 2.2. Drugs Targeting Hexokinase (HK)

The first rate-limiting enzyme of glycolysis is HK, which catalyzes the conversion of glucose to glucose-6-phosphate (G-6-P). Since G-6-P is a common intermediate product of glycolysis, PPP, and glycogen synthesis, this process is considered to be the most critical step in the process of glucose metabolism, and HK is regarded as the most important rate-limiting enzyme. There are four isoforms of mammalian HK named HK1, HK2, HK3, and HK4. HK1, HK2, and HK3 are high-affinity HKs, and HK1 and HK2 can specifically bind in mitochondria to voltage-dependent anion channels (VDACs), known as mitochondrial porins [99]. The autocatalytic product G-6-P mediates feedback inhibition of HK1, HK2, and HK3 activity, and G-6-P induces conformational changes in HK1 and HK2 that separate them from the mitochondria [99,100]. By binding to the outer mitochondrial membrane and VDAC, HK1 and HK2 preferentially dephosphorylate glucose using mitochondria-derived ATP, thereby linking OXPHOS and glycolysis [101]. HK1 is widely expressed in numerous organs, and HK2 expression is significantly upregulated in cancer cells, promoting glucose uptake and participation in multiple metabolic pathways [101]. Therefore, the high HK activity in cancer cells mainly results from the induced expression of HK2 [101]. In addition, p53 family members (p53, p63, and p73) play a significant role in regulating HK2 [102]. p53 can bind to the HK2 gene promoter, thus suppressing HK2 transcriptional activity and regulating its expression [103,104]. p63 and p73 are homologs of p53 and share some common functions with p53 [105,106]. However, p63 and p73 are more complex in structure, containing two major isoforms of each protein (TAp63, ΔNp63, TAp73, and ΔNp73). Similar to p53, TAp63 and TAp73 can inhibit glycolysis by inhibiting HK2 [107]. Contrary to TAp63, ΔNp63 was shown to upregulate the expression of HK2 [107,108]. Furthermore, the high expression of HK2 in cancer tissue cells is directly related to DNA methylation [100]. Overall, the elevated expression of HK2 causes significantly more efficient glycolysis in malignant tumors than in normal cells, which promotes the proliferation of cancer cells. HK2 is barely expressed in normal cells; therefore, its systematic knockdown selectively targets cancer cells [109]. Further studies revealed that germline knockdown of HK2 results in embryonic death, but systemic knockdown of HK2 in adult mice did not affect their survival. In addition, knockdown of HK2 was found to inhibit cancer development in mouse models and, more importantly, did not activate HK1 expression [109]. Several studies have shown that systemic inhibition of HK2 can safely and effectively block cancer growth [100,110,111,112]. However, due to the high structural similarity of HK1 and HK2 [113,114], the development of specific inhibitors remains a great challenge.

Many HK inhibitors have been exploited for anticancer effects. Among them, 2-deoxy-d-glucose (2-DG), lonidamine (LN), and 3-bromopyruvate (3-BrPA) have been the most studied. These molecules all target HK2 in many in vitro and in vivo tumor models, detach it from mitochondria, and elicit cancer cell death [112]. 2-DG (Figure 4A) is a glycolysis inhibitor that targets HK2 and competes with glucose for HK to inhibit glycolysis [115]. Preclinical studies have demonstrated that 2-DG significantly inhibits glycolysis and ATP synthesis [115]. Despite the promising results of 2-DG in preclinical studies, the results of clinical trials have been inconsistent [116,117,118]. Currently, 2-DG has been reintroduced for use in combination approaches, using 2-DG to produce synergistic anticancer effects with other anticancer agents [119,120]. In several clinical studies, 2-DG has been used as an adjuvant to clinical chemotherapeutic agents for various cancers, including breast, prostate, ovarian, lung, and glioma [121,122,123]. However, the use of 2-DG in cancer therapy is still limited. Studies have shown that the plasma half-life of 2-DG is only 48 min, and 2-DG must be administered at a relatively high concentration (5 mmol/L) to compete with blood glucose [121]. However, high doses of 2-DG can lead to adverse effects such as fatigue, sweating, dizziness, nausea, and hypoglycemia [123]. LN (Figure 4B) was previously used as an antispermogenic agent, but now it is known to have anticancer and proapoptotic effects [124]. LN is an adenine nucleotide translocator (ANT) ligand that induces mitochondrial channel formation and inhibits complex I and complex II [125]. As an emerging glucose metabolism enzyme-targeted drug, LN can be used alone or in combination with other anticancer agents. This agent has entered clinical trials for cancer treatment [126], such as lung, breast, and ovarian cancer [127,128,129,130]. Nevertheless, significant pancreatic and hepatic toxicities have limited LN’s clinical success [131]. Combination therapy studies revealed that combination with other chemotherapeutics, such as doxorubicin, produced better anticancer effects for the treatment of breast, prostate, and ovarian cancers [132,133]. To reduce LN toxicity, current research has focused on developing alternative dosage forms or local targeted delivery of LN. Nanomedicines for LN have been shown to inhibit glucose metabolism in cancer cells and regulate the immunosuppressive microenvironment, indicating great promise for the development of nanomedicines targeting glycolysis [134]. 3-BrPA (Figure 4C) is another HK2 inhibitor that can directly inhibit HK2 activity, thereby strongly inhibiting glycolysis [135]. 3-BrPA has been shown to enhance the cytotoxic effect and decrease resistance to other anticancer drugs by inhibiting the ATP-dependent multiple drug resistance (MDR) transporter, providing a promising candidate in combination therapy [136,137]. Regrettably, these molecules inhibit all HKs with less specificity for HK2, with the evident risk of suppressing glucose phosphorylation and utilization in crucial normal organs. Therefore, improving the pharmacokinetic properties of HK inhibitors, prolonging the half-life of the drug, synthesizing novel analogs or prodrugs of HK inhibitors, and enhancing the targeting of such drugs to cancer cells in vivo to reduce the occurrence of adverse effects may be essential strategies to break the limits of clinical application.

Many new HK inhibitors have been identified in recent years. For example, metformin can reduce mTORC1 activity in HCC cells, inhibiting protein synthesis and inducing cancer cell death in the absence of HK2 expression [112]. Several flavone derivatives, including oroxylin A (Figure 4D), chrysin (Figure 4E) [112], amentofavone (AF) (Figure 4F) [138], Gen-27 (Figure 4G) [139,140], and GL-V9 (Figure 4H) [138], have shown anticancer effects targeting HK2. Specifically, oroxylin A reduces HK2 expression and inhibits the binding of HK2 to mitochondrial VDAC, which is dependent on the deacetylation of procyclin D by SIRT3 [141]. Similarly, methyl jasmonate (MJ) (Figure 4I) can also inhibit HK2 expression and suppress HK2 and VDAC binding [142,143,144]. However, the selectivity of MJ to HK2 in cancers is relatively poor. Novel HK2 inhibitors, such as benserazide (Figure 4J) [145] and benitrobenrazide (Figure 4K) [146], have also shown effects in cancer therapy. However, the relevant studies are limited and further exploration is needed.

Many natural products or natural compounds, such as arsenic trioxide (ATO), curcumin, and epigallocatechin gallate (EGCG). have also been shown to inhibit HK2, suppressing growth and inducing apoptosis in cancer cells. Arsenic trioxide (ATO) (Figure 4L) is the main active ingredient of the traditional Chinese medicine (TCM) arsenic, which can inhibit the growth of gastric cancer by regulating glucose metabolism through downregulation of HK2 expression [147]. Curcumin also inhibits colorectal cancer growth by downregulating HK2 expression [148]. Epigallocatechin gallate (EGCG) dose-dependently inhibits the anchorage-independent growth of human tongue squamous cell carcinoma. It reduces HK2 protein expression by inhibiting the AKT pathway to suppress glycolysis and inhibits HK2 binding to mitochondria to promote apoptosis [149]. Dai et al. [150] found that resveratrol inhibited glycolysis and induced apoptosis in hepatocellular carcinoma cells by inhibiting HK2 expression to activate mitochondria-associated apoptotic signaling, and that it could also enhance the antihepatocarcinogenic effect of sorafenib. In addition, natural products or natural compounds such as bufalin (Figure 4M) [151], cryptotanshinone (Figure 4N) [152], revsveratrol (Figure 4O) [153], shikonin [154], fenofibrate [155], halofuginone [156], licochalcone A [157], jolkinolide B [158], ginsenoside 20(S)-Rg3 [159], ketoconazole [160], posaconazole [160], and astragalin [161] have also exhibited inhibitory effects on HK2.

### 2.3. Drugs Targeting Phosphofructokinase (PFK)

The second rate-limiting enzyme of glycolysis is PFK, which catalyzes the conversion of fructose-6-phosphate (F-6-P) to fructose-1,6-bisphosphate (F-2,6-BP). PFK is allosterically activated by adenosine monophosphate (AMP) and F-2,6-BP. PFK can be inhibited by the elevated F-2,6-BP level to sustain cancer cell growth [162]. F-2,6-BP, a product of the reaction catalyzed by 6-phosphofructo-2-kinase/fructose-2,6-bisphosphatase (PFK2/FBPase-2, PFKFB), is the most potent positive allosteric effector of PFK1 [163]. PFK2/FBPase-2 is a bifunctional enzyme responsible for the catalysis of both the synthesis and degradation of F-2,6-BP mediated through its N-terminal domain (2-Kase) and C-terminal domain (2-Pase), respectively [164]. In other words, PFKFB is an enzyme with both kinase and phosphatase activities, and the level of F-2,6-BP depends on the relative activities of kinase and phosphatase. Therefore, inhibiting the kinase activity of PFKFB while maintaining its phosphatase activity can inhibit PFK1 activity by reducing F-2,6-BP levels to block cancer growth [165]. In addition, PFKFB3 is commonly overexpressed in breast, colon, ovarian, and thyroid cancers but is expressed at low levels in normal tissues and is the basis of targeted therapy for a variety of cancers [166]. It was also found that inhibiting PFKFB3 could suppress pathological angiogenesis without affecting normal blood vessels.

PFK is controlled by a family of bifunctional enzymes, including PFKFBs [167]. PFKFB3 is overexpressed in various cancers, including breast, colon, nasopharyngeal, pancreatic, and gastric cancers, and is associated with lymph node metastasis and survival [168]. A large number of inhibitors of PFKFB3 have been reported, including 3PO (Figure 5A) [169,170], PFK15 (Figure 5B) [171], PFK158 (Figure 5C) [172,173], YN1 (Figure 5D) [174], and N4A (Figure 5E) [174]. In contrast to 2-DG, which can cause serious toxicity and systemic adverse effects, 3PO only partially and transiently reduces glycolysis without causing serious toxicity to normal tissues. Administration of 3PO was shown to produce a rapid reduction in glucose uptake, lactate production, and ATP generation in Jurkat T-cell leukemia cells [170]. PFK15, a derivative of 3PO, exhibits approximately a 100-fold increase in PFKFB3 inhibitory activity when compared to 3PO. PFK15 has been reported to exhibit significant anticancer activity by reducing 18FDG uptake and F-2,6-BP levels in xenografted tumors. Moreover, PFK15 exhibits a proapoptotic effect in transformed cancer cells in vivo and in vitro [171]. Several studies have demonstrated the ability of PFK15 and PFK158 to synergize with targeted and chemotherapeutic agents [173,175,176]. In addition, it has also been shown that PFK15 increases the sensitivity of chronic granulocytic leukemia cells to imatinib and enhances the cytotoxicity of oxaliplatin against colorectal cancers [177,178]. The combination of CTLA-4 antibody and PFK158 can significantly enhance the inhibition of cancer growth, showing a bright future for immunotherapy combined with targeted glucose metabolism therapy [179]. Some studies have found that PFKFB3 plays a key role in the repair of DNA damage by homologous recombination, leading to the development of the small-molecule PFKFB3 inhibitor KAN0438757 (Figure 5F), suggesting that PFKFB3 may play a key role in the initiation and development of malignant tumors [180]. Moreover, compounds such as BrAcNHEtOP (Figure 5G) [167], YZ9 (Figure 5H) [167], PQP (Figure 5I) [181], KAN0436151 (Figure 5J), benzindoles [182], and salicylic acid sulfonamides [183] have also been found to have pharmacological effects on PFK inhibition.

### 2.4. Drugs Targeting Pyruvate Kinase (PK)

The third rate-limiting step in glycolysis is pyruvate kinase (PK), which catalyzes the dephosphorylation of phosphoenolpyruvate to produce enolpyruvate. In mammalian cells, there are four main isoforms of PK: PKM1, PKM2, PKR, and PKL. PKL and PKR are mainly expressed in the liver and erythrocytes, respectively, and PKM1 is highly expressed in muscle and brain tissues. In cancer cells, low-affinity PKM2 is the main isoform. Further research has found that PKM2 exists in different forms, and in cancers, it mostly exists as a low-activity dimer, a form that is more likely to promote cancer growth [184]. In addition, PKM2 can exert its effects through post-translational modifications, including phosphorylation [185], O-acetylglucosamine (O-GlcNAc) modification [186], acetylation [187], succinylation [188] and methylation [189]. Numerous studies have shown that inhibition of PKM2 can improve the sensitivity of cancer cells to chemotherapeutic drugs such as cisplatin and reverse drug resistance [190,191]. Interestingly, either the inhibition or activation of PKM2 inhibited the growth of cancer cells, which may be related to the response of cancer cells to different degrees of hypoxia [192,193].

There are three main types of PKM2 inhibitors that have been identified: shikonin, metformin, and vitamin K. Shikonin (Figure 6A), the active ingredient extracted from the plant comfrey, is the most potent and specific PKM2 inhibitor reported to date. Shikonin’s analog, alkannin, also shows potential anticancer therapeutic value in targeting PKM2 [194]. Shikonin reduces platinum resistance in human colorectal and advanced bladder cancer cells by inhibiting PKM2 activity and reverses cisplatin resistance in cervical cancer cells in a dose-dependent manner [194]. In addition, shikonin significantly reduced gefitinib resistance in lung cancer cells and inhibited the development and metastasis of esophageal and bladder cancers [195,196]. However, due to shikonin’s poor solubility and complex pharmacological activity [197], there are still many safety concerns for its direct incorporation into treatment protocols. Therefore, optimizing the drug structure to target and enhance anticancer activity, utilizing nanoformulations, and other methods to enhance drug solubility are ways to overcome these limitations. In the past decade, various advanced drug delivery systems have been widely reported, including nanoparticles [198,199], liposomes [200,201,202], microcapsules [203], electrospun nanofibres [204], microemulsions [205], microneedles [206], polymeric micelles [207], etc. These nano-delivery systems transcend the limitations of conventional carrier systems and facilitate the precise delivery of shikonin and its derivatives to the target site of action [208,209,210]. Metformin (Figure 6B) is a commonly recommended drug for type II diabetes mellitus, but several studies have shown that metformin also has high potential as an anticancer agent [211]. Metformin enhanced the sensitivity of osteosarcoma stem cells to cisplatin by decreasing the expression level of PKM2 and inhibited glucose uptake, lactate production, and ATP production in osteosarcoma stem cells [212]. In addition, the combination of metformin and anti-ENO1 antibody significantly reduced the resistance of human non-small cell lung cancer cells to cetuximab and activated AMPK to downregulate PKM2 to inhibit metastasis and invasion of kidney cancer cells [213]. However, metformin is not highly selective for PKM2, and its pharmacological effects and clinical applications in other fields are very complicated. Therefore, the clinical use of metformin for cancer therapy needs further exploration. Vitamin K (VK) is a fat-soluble naphthoquinone, of which the VK3 (Figure 6C) and VK5 (Figure 6D) isoforms can inhibit PKM2 with an inhibitory effect that is more significant than that of PKM1 [214]. Studies have shown that the combination of VK3 and vitamin C (VC) can improve the therapeutic effect [215] and clinical trials have shown that VK3 can reverse cellular resistance to doxorubicin and adriamycin [216]. However, the clinical use of VK as an adjuvant for reversal of drug resistance is limited, which may be related to the contraindication of the use of VK for hepatic dysfunction, as patients with cancers treated with long-term chemotherapy are prone to impaired liver function or hepatic dysfunction, which essentially limits the application of VK in anticancer therapy. Zhou Y et al. [217] recently discovered that benserazide (Figure 6J), a dopa decarboxylase inhibitor for Parkinson’s disease, was also able to specifically bind and block PKM2 enzyme activity and inhibit glycolysis, which inhibited the growth of melanoma. Such discoveries provide additional ideas for drug combinations for the treatment of cancers. In addition, several compounds, such as lapachol (Figure 6E) [218], C3k (Figure 6F) [219], benzoxepane derivatives (Figure 6G) [220], cyclosporin A (CsA), tannic acid (TA), and beta-elemeneand can inhibit PKM2, leading to the suppression of glycolysis in cancer cells.

Several natural products or natural compounds have also been shown to inhibit HK2. For example, Liu et al. [221] found that oleanolic acid (OA) induced the conversion of PKM2 to PKM1 and attenuated the Warburg effect, suggesting that OA is a compound that inhibits aerobic glycolysis. MC-4, an extract from Artemisia annua, reduced the expression of PKM2 and GLUT1 and significantly inhibited cancer growth [222]. Further research found that the combination of MC-4 and everolimus can synergistically exert anticancer effects through AKT/PKM2 and mTOR to inhibit cancer growth and metastasis, which provides a theoretical basis for the combination of targeted therapy with glycolysis inhibitors [222]. In addition, many natural products or natural compounds, such as curcumin [223,224], resveratrol [225,226], proanthocyanidin B2 (PB2) (Figure 6H) [227], apigenin, wogonin, chysin, and many more [228,229] are also able to bind to the variable site of PKM2 and inhibit glycolysis.

As mentioned earlier, most PKM2 in cancer tissues are low-activity dimers, which catalyze relatively less pyruvate production, thus providing sufficient intermediate components for conversion into proteins, nucleotides, and other vital substances necessary for cancer cell proliferation [229]. In a breast cancer model, knockdown of PKM2 can enhance tumor formation, suggesting that PKM2 inhibition alone may not be effective [230]. TEPP-46 and DASA-58, both PKM2 activators, significantly increased the level of highly active PKM2 tetramers, which hindered tumorigenesis in animal experiments, inhibited the metabolism of nucleotides and serine and reduced lactate production [231]. Mohammad et al. [232] found that the use of TEPP-46 significantly enhanced PK activity in pancreatic cancer cells, downregulated PKM2 dimer expression, and inhibited the growth of tumors in a mouse model. In addition, many molecules, such as parthenolide (PTL 5) (Figure 6I) [233], ML-265 (Figure 6K) [234], PA-12 [235], Pyridin-3 ylmethyl carbamodithioic esters [236], ZINC08383544 [237], and compound 0089-0022 [238], inhibit tumor growth as PKM2 agonists. Some researchers have also explored the effects of herbal components on PKM2 enzyme activity. Aslan et al. [239] found that polyphenolic extracts such as prunetinone and quercetin flavonoids have efficient activating effects on PKM2 enzyme activity. Mustard acid and p-coumaric acid can also act as PKM2 activators for anticancer effects [239].

Although many studies have been conducted on inhibitors and activators targeting PKM2, their specific applications have not been fully explored. For example, in individualized cancer therapy, it is worthwhile to investigate which types of cancer and at which stage of cancer the inhibitors and activators should be used. With development and research of activators and inhibitors with high specificity, drugs targeting PKM2 will become more widely used in cancer treatment.

### 2.5. Drugs Targeting Lactate Dehydrogenase (LDH)

Lactate dehydrogenase (LDH) catalyzes the last step in the glucose metabolism process, catalyzing the reversible conversion of pyruvate to lactate. The accumulation of lactate affects the pH in the TME. As a proinflammatory and immunosuppressive mediator, lactate promotes the malignant progression of tumors. Studies have shown that high levels of lactate are associated with early distant metastasis of cancer [240]. Lactate also activates matrix metalloproteinase (MMP) histone proteases; upregulates vascular endothelial growth factor (VEGF), HIF-1α, and transforming growth factor-β2 (TGF-β2); and directly enhances the migration ability of cells [241]. The human genome has four LDH genes: LDHA, LDHB, LDHC, and LDHD. LDHA and LDHB are highly expressed in cancers [242], with LDHA responsible for converting pyruvate to lactate and LDHB responsible for converting lactate to pyruvate. The predominant isoform in cancers is LDHA [23]. High LDHA expression is related to the poor prognosis of malignant tumors [243,244]. In addition, LDHA can also promote lactate production, thereby remodeling the TME and suppressing the immune system to promote immune escape [23,65]. Furthermore, it was found that upregulation of LDHA ensures efficient aerobic glycolysis in cancer cells, but the enzyme is not required for healthy cells under normal conditions [245]. Knockdown of LDHA can inhibit cancer cell proliferation, suggesting that targeting LDHA is a promising strategy to inhibit the growth of malignant tumors. In addition, knockdown of LDHA can also have an effect on matrix metalloproteinases, thus affecting cancer cell invasion and metastasis [246]. Therefore, the main target for developing anticancer drugs against LDH is LDHA.

There is currently much research devoted to the search for selective inhibitors of LDHA [23]. The natural compound gossypol (Figure 7A), a nonselective LDHA inhibitor, has shown efficient anticancer activity in vitro and in preclinical experiments, but gossypol also interacts with other components in the cell involving multiple biological functions, leading to nonspecific toxicities [247]. Therefore, there is a need to design chemically synthesized LDHA inhibitors to improve the efficiency and safety of these drugs. FXII (Figure 7B), a catechol-containing small compound that inhibits LDHA, was shown to inhibit tumor growth in xenografts [248]. In lymphoma and pancreatic cancer, FX-11 can reduce cellular lactate production, induce oxidative stress, and ultimately lead to apoptosis and the inhibition of cancer progression [248]. In prostate cancer, FX-11 as a single agent was also shown to be effective in inhibiting the glycolysis of cancer cells and consequently the growth of cancer cells [248]. In addition, galloflavin (Figure 7C) has been reported to bind to free LDHA and inhibit glycolysis in breast cancer cells, which inhibits cancer growth [249]. Oxamate (Figure 7D) is a competitive LDH inhibitor that exerts its pharmacological effects by competing with the LDHA substrate pyruvate. When combined with LDHA, oxamate can inhibit the conversion of pyruvate to lactate by inhibiting LDH and inhibiting the proliferation and migration of prostate and breast cancer [250,251,252], and its sensitivity can be effectively improved when combined with temozolomide [253]. However, in vitro studies have shown that oxalate requires concentrations above the millimolar level to exert anticancer effects. Notably, oxidative cancer cells are less sensitive to LDHA inhibitors, while some glycolytic cancer cells will compensate for the inhibition of glycolysis by OXPHOS and become resistant to LDHA inhibitors. Therefore, LDHA inhibitors can be used in combination with OXPHOS inhibitors (e.g., phenylephrine) to exert a more comprehensive anticancer effect [254]. In addition, morin (Figure 7E), EGCG (Figure 7F), the NADH competitive inhibitor GSK2837808A [255], pyruvate and NADH competitive inhibitors NHI1 and NHI2 [256], metamorphic inhibitor PSTMB [257], and piperidine derivative GNE140 [258] all have strong inhibitory and selective effects on LDHA and can inhibit cancer progression with less effect on normal cells, and are therefore considered as potential novel anticancer drugs [259].

### 2.6. Drugs Targeting Aldolase (ALDO)

Aldolase (ALDO) catalyzes the breakdown of F-1,6-BP to dihydroxyacetone phosphate and glyceraldehyde-3-phosphate in a reversible reaction. ALDO can bind to actin fibers, and the PI3K signaling pathway allows ALDO to separate from actin fibers and promote glycolysis [260]. ALDOA expression is significantly increased in HCC tissues and is associated with the malignant progression of HCC [261]. On the other hand, dietary restriction (DR) can upregulate the ALDOA/DNA-PK/p53 pathway, a potential mechanism for the anticancer effect of DR [262]. TDZD-8, a small molecule metamorphic inhibitor, has been found to specifically target the Cys289 site of ALDOA, inhibit the glycolytic function of ALDOA, and reduce the stability of HIF-1α to exert anticancer effects [263]. In addition, Chang et al. [264] found that raltegravir suppresses lung cancer metastasis by inhibiting ALDOA–γ-actin interactions and was not significantly toxic to normal lung tissue. However, this was limited to laboratory studies, and raltegravir is mainly useful in antiviral therapy, so the therapeutic role of raltegravir in tumors needs to be further explored.

### 2.7. Drugs Targeting Phosphoglycerate Kinase 1 (PGK1)

Another key enzyme in glycolysis is phosphoglycerate kinase 1 (PGK1). In non-small cell lung cancer cells, the long noncoding RNA MetaLnc9 interacts with the glycolytic kinase PGK1. It prevents ubiquitination, which activates the oncogenic AKT/mTOR signaling pathway and accelerates cancer progression [265]. Overexpression of the proto-oncogene gankyrin attenuates cellular oxidative stress and increases the oncogenic properties of gastric cancer cells through activation of the PGK1/AKT/mTOR pathway [266]. However, since there is a lack of promising lead compounds, studies on PGK1 inhibitors are comparatively weak [267]. Moreover, the effects of existing PGK1 inhibitors such as CBR-470-1, bisphosphonates, terazosin, and their derivatives on cancer cells have not been reported [268].

### 2.8. Drugs Targeting Phosphoglycerate Mutase 1 (PGAM1)

Phosphoglycerate mutase 1 (PGAM1), which is regulated by TP53, is commonly upregulated in human cancers and promotes cancer cell proliferation and cancer growth by regulating the levels of its substrate 3-PG and product 2-PG [269]. Moreover, the expression level of PGAM1 was negatively correlated with the prognosis of cancer patients and positively correlated with tumor stage and pathological grade in HCC [270], bladder cancer [271], and lung cancer [272]. Therefore, PGAM1 is a promising target for antitumor drugs. Evans et al. [273] first reported the small molecule compound MJE3, which can specifically act on PGAM1 and inhibit the proliferation of breast cancer cells. Hitosugi et al. [274] identified three compounds through in vitro screening and obtained PGMI-004 after structural optimization; it can selectively inhibit PGAM1 activity, significantly inhibit glycolysis and PP in cancer cells, and reduce the synthesis of biomolecules such as nucleotides, amino acids, and lipids, while being less toxic to normal cells.

### 2.9. Drugs Targeting Enolase (ENO)

Enolase (ENO) catalyzes the reversible reaction of phosphoenolpyruvate production and is highly expressed in nasopharyngeal carcinoma and non-small cell lung cancer. ENO can promote cell proliferation, migration, and invasion by upregulating glycolysis through activation of the PI3K/AKT pathway [275]. The expression of ENO1 is elevated in several cancer tissues, suggesting its close association with carcinogenesis. According to Yin H et al. [276], ENO1 overexpression in pancreatic cancer is associated with clinical stage, lymph node metastasis, and poor prognosis. ENO1 also promotes cisplatin resistance in patients with gastric cancer [277]. Chemical enolase inhibitors include sodium fluoride, D-tartonate, and 3-aminoenolpyruvate 2-phosphate, but none of these are appropriate for cancer therapy [278,279]. Phosphonoacetohydroxamicacid (PhAH), a pan-enolase transition-state analogue inhibitor, can inhibit both enzymatic activity and proliferation in cancer cells, including pancreatic, breast, and lung cancers [280,281]. In addition, ENOblock (AP-III-a4) has also been found to have anticancer effects [282]. Overall, concerted efforts are still required to develop suitable drugs that do not affect normal cells.

### 2.10. Drugs Targeting Monocarboxylate Transporters (MCTs)

In addition to the above glucose metabolism enzymes, monocarboxylate transporters (MCTs) are also functional molecules essential for the glycolytic process and play a vital role in the growth of cancer cells. MCTs are responsible for transporting lactate produced by glycolysis to the extracellular compartment, preventing excessive acidification of the cytoplasm, and protecting cells from damage caused by the acidic environment. MCT overexpression in cancer cells can maintain the appropriate pH for cancer growth, thus promoting proliferation [283]. AstraZeneca developed AZD3965, a selective inhibitor of MCT1 which was shown to inhibit the bidirectional transport of lactate in cancers. AZD3965 caused an increase in intracellular lactate content and a decrease in ATP, which in combination with radiotherapy reduced cancer growth and prolonged survival, enhancing radiotherapy sensitivity [284]. A recent study found that the MCT inhibitor AZD3965 also inhibits lipid biosynthesis and increases tumor immune cell infiltration involving dendritic cells (DCs) and NK cells in the TME [285]. AZD3965 in combination with an anti-PD-1 antibody reverses the immunosuppressive microenvironment of solid tumors by targeting MCT1.

A nanodrug composed of an MCT1 inhibitor (AZD3965) loaded inside the ultra-pH-sensitive nanoparticles (AZD-UPS NPs) can reduce the dose of AZD3965 and can increase the effect of immunotherapy [286]. AZ93 has been reported to selectively inhibit MCT4 and has been used in preclinical studies [287]. More compounds that can effectively reduce lactate flux are 7-aminocarboxycoumarins (7ACCs). 7ACCs retard the growth of a variety of cancer cells, and 7ACCs inhibit the recurrence of cervical cancer after cisplatin treatment [288]. In addition, MCT inhibitors such as AR-C155858 [289] and VB124 [290] have also been found to have an anticancer effect.

### 2.11. Drugs Targeting Isocitrate Dehydrogenase (IDH)

IDH is a family of metabolic enzymes with important roles in the TAC cycle that is widely involved in glucose metabolism, amino acid metabolism, and lipid metabolism [291]. The main role of IDH is to catalyze the oxidative decarboxylation of isocitrate to generate a-ketoglutarate (a-KG), while reducing nicotinamide adenine dinucleotide (NAD^+^) and beta-nicotinamide adenine dinucleotide phosphoric acid (NADP^+^) to the reduced form of nicotinamide adenine dinucleotide (NADH) and the reduced form of nicotinamide adenine dinucleotide phosphate (NADPH). IDH1/2 mutations promote the development of various cancers, such as lymphoma and glioma; ivosidenib, which targets mutated IDH1, and enasidenib, which targets IDH2, were approved for marketing and use in the treatment of acute myeloid leukemia in 2017 and 2018, respectively [292,293,294]. Moreover, other compounds, such as IDH305 and AG-881, have been in clinical trials [295].

## 3. Combinational Strategies Using Glucose Metabolism Enzyme Inhibitors

Despite the rapid development of various small-molecule inhibitors of glucose metabolism enzymes, their clinical applications are still limited. Combination with other anticancer drugs may show enhanced anticancer effects. For example, the combination of PKM2 activators and LDHA inhibitors significantly reduced cancer growth in a mouse model of pancreatic adenocarcinoma transplantation, suggesting the potential value of multitarget glycolytic inhibitors in combination [232]. In addition, treatment with drugs targeting glucose metabolism enzymes will cause a compensatory enhancement in the metabolism of other nutrients in cancer cells [254]. For example, significantly elevated levels of redox reactions were found in various cancer cells after glycolysis was inhibited [296,297]. This problem can be solved by combining a glucose metabolism enzyme inhibitor with an OXPHOS inhibitor [254,298].

In addition, increasing evidence suggests that aerobic glycolysis not only promotes cancer cell proliferation but is also associated with chemotherapy resistance. Therefore, drugs targeting glycolysis may provide additional killing capacity for chemotherapy. For example, Korga et al. [299] used 2-DG in combination with adriamycin to treat liver cancer cells. The results showed that the combination therapy was more effective in inhibiting liver cancer cell activity and promoting apoptosis than adriamycin treatment alone. Further studies revealed that 2-DG inhibited protein N-glycosylation and improved the efficacy of standard chemotherapy through chemo-sensitization and reversing resistance to 5-fluorouracil (5-FU) in prostate cancer cells, trastuzumab in breast cancer cells, and Bcl-2 inhibitors in leukemia cells [300,301]. The use of 2-DG also significantly reduced resistance to paclitaxel and adriamycin in osteosarcoma and non-small cell lung cancer transplanted mice when compared with chemotherapy alone [302]. 2-DG combined with sorafenib and 2-aminophenoxazine-3-one (Phx-3) also enhanced the anticancer effect of 2-DG in hepatocellular carcinoma [115,303]. The GLUT1 inhibitor BAY876 was found to enhance cisplatin-mediated antiproliferative effects in laryngeal squamous carcinoma [304]. This suggests that glycolysis inhibitors can enhance the sensitivity of cancer cells to chemotherapeutic drugs. The mechanism by which this occurs seems to be that glycolysis inhibitors deprive the energy supply of cancer cells, thus reducing the resistance of cancer cells to chemotherapeutic drugs. In conclusion, combining chemotherapeutic agents and glycolysis inhibitors is a promising strategy for the treatment of cancer.

A negative correlation was found between glucose metabolism enrichment scores and immune cell activity in triple-negative breast cancer. It is suggested that combining PD-1 or PD-L1 antibodies with glycolytic inhibitors is a promising therapeutic strategy. The combination of a PD-1 inhibitor and LDH inhibitor FX11 significantly increased tumor CD8^+^ and NK cells infiltration and demonstrated remarkable anticancer effects [305]. Zappasodi et al. [306] found that knockdown of LDH and blockade of CTLA-4 in a high glycolytic mouse breast cancer tumor model promoted immune cell infiltration and Treg cells were forced to participate in glycolysis in the presence of glucose, enhancing glucose uptake and IFN-γ production, leading to a loss of Treg cell stability. Blocking CTLA-4 is more suitable for treating cancers with low levels of glycolysis, while for cancers with high levels of glycolysis, the combination of anti-CTLA-4 antibodies with glycolysis inhibitors increases the availability of glucose in the TME, which maximizes Treg cell instability and enhances anticancer immunity [306]. Of note, diclofenac reduced lactate secretion and enhanced the killing ability of infiltrating T cells in in vitro experiments [307]. Diclofenac has previously been shown to be an MCT1/4 inhibitor, and studies support the concept of combining glycolytic inhibitors and immune checkpoint inhibitors in clinical trials for the treatment of highly glycolytic cancers. Ho et al. [308] found that tumor-infiltrating T cells compete with cancer cells for metabolism in the TME. Reducing the glucose level in the TME can inhibit the response of infiltrating T cells against cancer cells. In addition, T cells express PD-1 on their surface, while cancer cells express the PD-L1 on their surface, which can escape T-cell immune surveillance. A recent study [290] found that the combination of MCT inhibitors and PD-1 monoclonal antibodies reduced the growth of transplanted tumors and increased the infiltration of CD8^+^ T cells in murine hepatocellular carcinoma and was able to significantly enhance the anticancer effects of anti-PD-1 antibodies. The above studies indicate a bright future for the combination of immune checkpoint inhibitors and glycolysis inhibitors.

## 4. Limitations of Drugs Targeting Glucose Metabolism Enzymes

Although preclinical studies have demonstrated the effectiveness of glucose metabolism enzyme-targeted anticancer drugs, their clinical translation has remained limited to date (Table 1). Overall, there are three main limitations in developing glucose metabolism enzyme-targeted therapies. Firstly, a key challenge in developing small-molecule inhibitors is that most of the key enzymes of glucose metabolism exist in multiple isoforms, and the structures of the different isoforms are highly similar. The low selectivity of targeted drugs leads to the occurrence of adverse effects. In addition, targeted drugs may cause compensatory activation of other isoforms in the tumor, thus reducing the efficacy of the agents. Although several small-molecule targeted drugs have been demonstrated effective or even entered clinical trials [298], the poor targeting will produce toxic side effects, making it difficult to meet cancer treatment requirements [116,117,118]. Therefore, the development of highly selective inhibitors targeting glucose metabolism enzymes remains a challenging endeavor. To date, most gene sequences and protein structures of glucose metabolism enzymes have been annotated. Therefore, it is possible to design small-molecule inhibitors based on the crystal structure of glucose metabolism enzymes using computer-assisted drug design. The analysis of the binding sites in the crystal structure of glucose metabolism enzymes and their interactions with substrates, the clarification of the relevant properties of the binding sites, and the identification of key binding residues and possible binding regions will facilitate the research of structure-based small-molecule drug design or structure modification. We hope that researchers will combine crystal structure docking studies for targeted small-molecule drug design of glucose metabolism enzymes. Additionally, chemical structure optimization guided by pharmacophore modeling and traditional medicinal chemistry design ideas will eventually lead to the construction of a new class of active small-molecule compounds for cancer treatment.

Hypoxia is a prominent feature of the TME, but there is significant heterogeneity in metabolic patterns across different cancer cells. Cancer cells close to blood vessels are mainly metabolized by OXPHOS. It has been found that cancer cells close to blood vessels can take up lactate via MCT1 and use it for tricarboxylic acid cycle energy supply [309]. In contrast, cancer cells distant from blood vessels take up glucose for glycolytic energy supply and release lactate. Such cancer cells with different metabolic patterns exhibit a phenomenon known as metabolic symbiosis, which makes cancer cells more adaptable to the harsh TME [23]. Therefore, a single glycolytic targeting drug cannot destroy cancer cells with metabolic heterogeneity. Instead, the metabolic stress induced by targeted drugs may promote the metabolic reprogramming of cancer cells, such as a greater reliance on glutamine metabolism, thus causing drug resistance. One potentially effective strategy is to treat the metabolic patterns of different cellular subpopulations in the TME to render them relatively homogeneous and then target this relatively homogeneous metabolic population to achieve disruption. For example, antiangiogenic drugs are given before glucose metabolism enzyme-targeted drugs to make the cancer more dependent on glycolysis and thus achieve better therapeutic results. Chaturvedi B et al. [310] treated melanoma by inhibiting the mitochondrial respiration of cells with metformin, making the cells dependent on glycolysis before using LDH inhibitors and achieving better therapeutic results.

Another challenge for glucose metabolism enzyme-targeted therapies is the immune system’s response to the drug. Several anticancer immune cells are dependent on glycolysis for their function. For example, cytotoxic T lymphocytes require an adequate supply of glucose to produce gamma interferon for their anticancer effects; NK cells’ activation depends on glycolysis, and restriction of glycolysis causes the depletion of NK cells; DCs depend on glycolysis for IL-12 production and promotion of T-cell proliferation; Th1 and Th17 cells require glycolysis for differentiation; and macrophage secretion of tumor necrosis factor (TNF) is glycolysis dependent. In addition, M1 polarization of macrophages also depends on glycolysis [311,312,313]. Overall, targeted inhibition of glucose metabolism enzymes can inhibit the growth of cancer cells but also suppress the anticancer immune response. The regulation pattern of cancer cell metabolism is significantly different from that of other cells, and if the differences in metabolic regulation in cancer cells and immune cells can be identified, targeted therapies addressing these differences have the potential to solve the above problems.

## 5. Conclusions

Numerous studies have demonstrated that tumorigenesis and metastasis development are closely related to the metabolic reprogramming of cancer cells. Small-molecule inhibitors acting on key enzymes of glucose metabolism can regulate cancer metabolic reprogramming to inhibit cancer cell growth. Studies on some glucose metabolism modulators, such as 3-BrPA, LN, and 2-DG, have been conducted for decades and have shown significant inhibition in various cancers. However, due to side effects, most drugs have failed to enter the clinic. Metformin, VK, and other drugs have been widely used in other fields, and their safety and efficacy are guaranteed. However, further research and clinical trials of their use in anticancer therapy are needed. Other small-molecule inhibitors of key enzymes, such as oxalate and salicylate sulfonamides, are still in their infancy, yet they have shown great potential for cancer treatment. In addition, some natural products have been identified to inhibit cancer cell growth by regulating key aerobic glycolysis enzymes, providing new ideas and strategies for developing anticancer drugs targeting glycolytic enzymes. However, the specific mechanisms of effect and targets are still unclear and need further investigation. Overall, developing anticancer drugs targeting glucose metabolism enzymes remains a significant challenge.

Most cancer cells have abnormal glucose metabolism, and the Warburg effect brings a new perspective to cancer treatment strategies. In this review, we have outlined the regulatory mode of glycolysis in cancer cells and presented the regulatory mechanism of GLUT, HK, PFK, PK, LDH, and other transporters or metabolic enzymes as targets in cancers and developed target drugs. Due to the unique metabolic features of cancer cells, the development and clinical translation of targeted therapeutic agents should be strengthened. Targeted glucose metabolizing enzyme drugs have been shown to have efficient anticancer effects in a variety of tumor models. Although no single glucose metabolism modulator is currently used in first-line clinical cancer treatment, combining glucose metabolism modulators with conventional anticancer drugs may become a promising cancer treatment strategy. Therefore, subsequent studies can not only explore the prognostic effects of glycolytic enzyme inhibitors on cancer patients but also accelerate the exploration of the combined application of glucose metabolism enzyme inhibitors and other anticancer drugs and translate the results into clinical treatment. With the development of new technologies such as high-throughput multi-omics and spatial omics, the heterogeneity of cancer cells and immune cells will be further elucidated, and therapeutic drugs targeting the glucose metabolism of malignant tumors will become an essential complement to existing treatments, thus changing the current state of cancer therapy. 

## Figures and Tables

**Figure 1 cancers-14-04568-f001:**
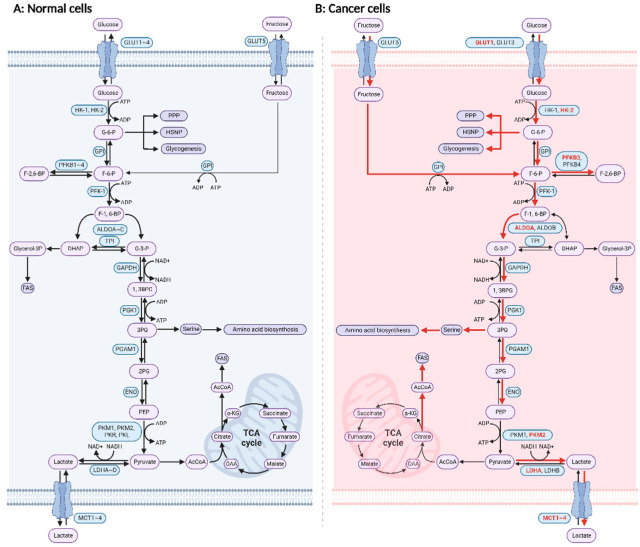
Reprogramming of glucose metabolism in cancer cells. (**A**) Glucose metabolism in normal cells; (**B**) glucose metabolism in cancer cells. Cellular uptake of glucose is followed by a series of reactions to transform glucose to pyruvate. Then, glucose enters the TCA cycle or is converted to lactate. Enzymes or pathways predominant in cancer cells are shown in bold red. Created with BioRender.com (accessed on 22 August 2022). Abbreviations: 1,3 BPG, 1,3-bisphosphoglycerate; 2-PG, 2-phosphoglycerate; 3-PG, 3-phosphoglycerate; α-KG, α-ketoglutarate; AcCoA, acetyl coenzyme A; ADP, adenosine diphosphate; ALDO, aldolase; ATP, adenosine triphosphate; DHAP, dihydroxyacetone-phosphate; ENO, enolase; F-1,6-BP, fructose-1,6-bisphosphate; F-2,6-BP, fructose-2,6-bisphosphate; F-6-P, fructose-6-phosphate; FAS, fatty acid synthesis; G-3-P, glyceraldehyde-3-phosphate; G-6-P, glucose-6-phosphate; HK, hexokinase; LDH, lactate dehydrogenase; GAPDH, glyceraldehyde-3-phosphate dehydrogenase; GCK, glucokinase; GLUT, glucose transporter; glycerol-3P, glycerol-3-phosphate; GPI, glucose-6-phosphate isomerase; MCT, monocarboxylate transporter; OAA, oxaloacetate; PEP, phosphoenolpyruvate; PFK1, phosphofructokinase 1; PFKFB, 6-phosphofructo 2-kinase/fructose-2,6-bisphosphatase; PGAM1, phosphoglycerate mutase 1; PGK1, phosphoglycerate kinase 1; PK, pyruvate kinase; PPP, pentose phosphate pathway; TCA, tricarboxylic acid; TPI, triosephosphate isomerase.

**Figure 2 cancers-14-04568-f002:**
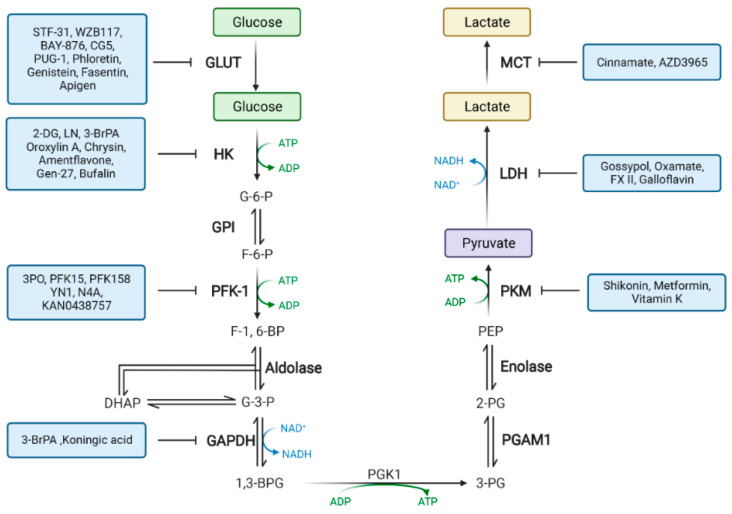
Drugs targeting glucose metabolism in cancer cells. Reprogramming of glucose metabolism provides many potential targets for cancer therapy. Created with BioRender.com (accessed on 22 August 2022). Abbreviations: 1,3 BPG, 1,3-bisphosphoglycerate; 2-PG, 2-phosphoglycerate; 3-PG, 3-phosphoglycerate; ADP, adenosine diphosphate; ALDO, aldolase; ATP, adenosine triphosphate; DHAP, dihydroxyacetone-phosphate; ENO, enolase; F-1,6-BP, fructose-1,6-bisphosphate; F-2,6-BP, fructose-2,6-bisphosphate; F-6-P, fructose-6-phosphate; G-3-P, glyceraldehyde-3-phosphate; G-6-P, glucose-6-phosphate; HK, hexokinase; LDH, lactate dehydrogenase; GAPDH, glyceraldehyde-3-phosphate dehydrogenase; GLUT, glucose transporter; GPI, glucose-6-phosphate isomerase; MCT, monocarboxylate transporter; PEP, phosphoenolpyruvate; PFK1, phosphofructokinase-1; PFKFB, 6-phosphofructo 2-kinase/fructose-2,6-bisphosphatase; PGAM1, phosphoglycerate mutase 1; PGK1, phosphoglycerate kinase 1; PK, pyruvate kinase; TPI, triosephosphate isomerase.

**Figure 3 cancers-14-04568-f003:**
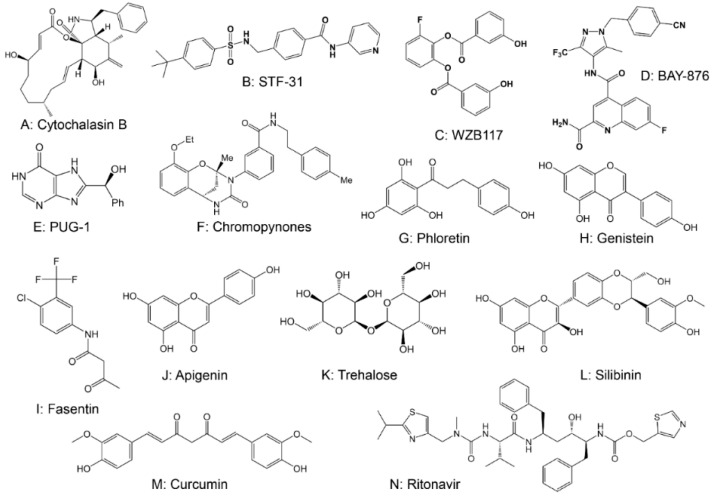
The chemical structures of drugs targeting glucose transferase.

**Figure 4 cancers-14-04568-f004:**
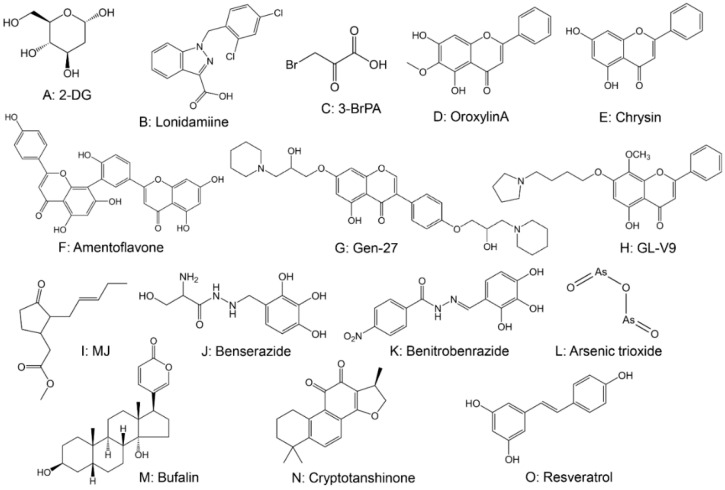
The chemical structures of drugs targeting hexokinase.

**Figure 5 cancers-14-04568-f005:**
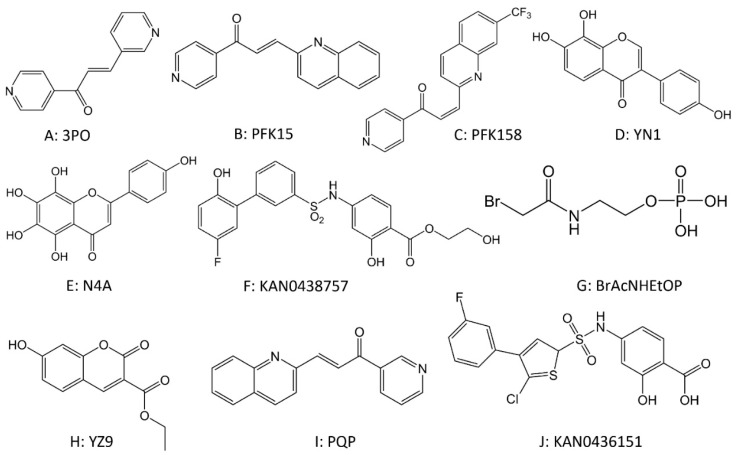
The chemical structures of drugs targeting phosphofructokinase.

**Figure 6 cancers-14-04568-f006:**
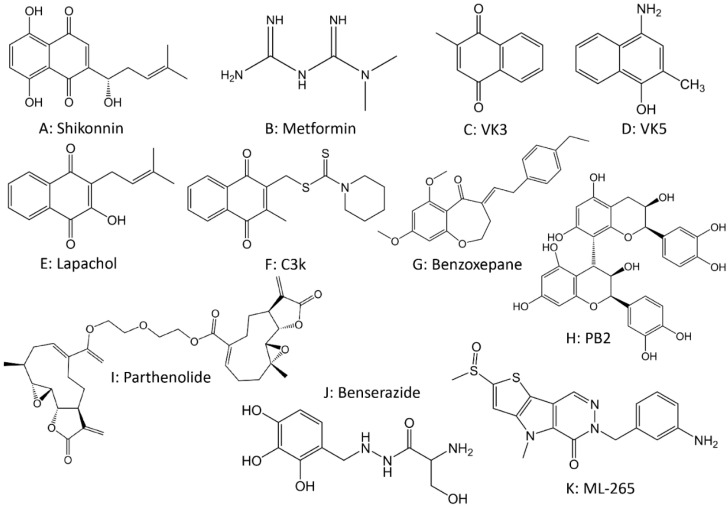
The chemical structures of drugs targeting pyruvate kinase.

**Figure 7 cancers-14-04568-f007:**
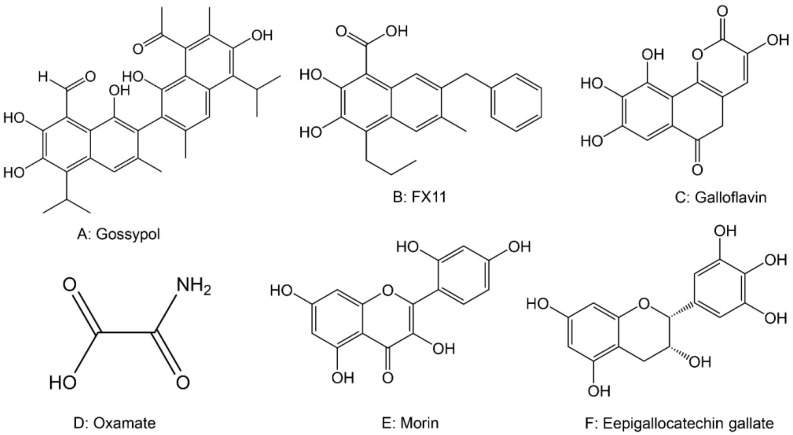
The chemical structures of drugs targeting lactate dehydrogenase.

**Table 1 cancers-14-04568-t001:** Main targets and drugs of glucose metabolism of cancer cells.

Enzyme	Target	Agents	Tumor Type	Study Phase
GLUT	GLUT1	STF-31	RCC	Preclinical
		WZB115, WZB117	BC, LC	Preclinical
		Fasentin	PC, Lymphoma	Preclinical
	GLUT1/2	Phloretin	HCC, BC, PC, LC, CC, etc.	Preclinical
	GLUT4	Ritonavir	Multiple myeloma, BC, CLL, etc.	Phase I/II clinical trial
	GLUT5	2,5-AM	Acute myeloid leukemia	Preclinical
HK	HK1/2	Lonidamine	HCC, BC, LC, Melanoma, OC, etc.	Phase I/II clinical trial
	HK2	2-DG	PC	Phase I/II clinical trial
		3-BrPA	HCC, BC, Pancreatic cancer, etc.	Phase I/II clinical trial
PFK	PFKB3	3PO	LC, Pancreatic cancer, etc.	Phase I clinical trial
		PFK15	RCC, HCC, CC, Gastric Cancer, etc.	Phase I clinical trial
		PFK158	LC, OC, etc.	Phase I clinical trial
PK	PKM2	Shikonin	BC, Skin cancer, Bladder cancer	Preclinical
		Orlistat	OC	Preclinical
LDH	LDHA	AT-101	Chronic lymphoblastic leukemia	Phase I/II clinical trial
		Glloflavin	BC	Preclinical
		Polyphenon E	BC, Colon cancer	Phase I/II clinical trial

Abbreviations: 2,5-AM, 2,5-anhydro-d-mannitol; 2-DG, 2-deoxy-d-glucose; 3-BrPA, 3-bromopyruvate; BC, breast cancer; CC, colorectal cancer; GLUT, glucose transferase; HCC, hepatocellular carcinoma; HK, hexokinase; LC, lung cancer; LDH, lactate dehydrogenase; OC, ovarian cancers; PC, Prostate cancer; PFK, phosphofructokinase; PK, pyruvate kinase; RCC, renal cell carcinoma.

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
