# Peer review of "Targeting Glucose Metabolism Enzymes in Cancer Treatment: Current and Emerging Strategies"

_cancers, 2022, doi:10.3390/cancers14194568_

Round 1

Reviewer 1 Report

I found this review complete and updated. A quick search on PubMed about this area shows a field that is still growing and this review provides a good compile of the current emerging treatments that target glucose metabolism.

I will accept this review in its present form. 

Author Response

Dear reviewer,

On behalf of all the contributing authors, I would like to express our sincere appreciation for your comments concerning our manuscript entitled "Targeting glucose metabolism enzymes in cancer treatment: current and emerging strategies" (Manuscript ID: cancers-1904543). Thank you for your positive comments and endorsement of our review. We have revised the manuscript according to the editor and another reviewer's comments.

Sincerely yours,

Bingwen Zou M.D.

Sep 14th, 2022

Reviewer 2 Report

General comment

Overall, this is a concise review that the regulatory mode of glycolysis in cancer cells and introduced transporters or metabolic enzymes as potential targets in cancers. Combination strategies with other anticancer drugs and the associated limitations were included. Overall, this is a high quality manuscript featured the significance of glucose metabolism regulator agents in new anticancer therapies. Specific comments are as below.

Simple Summary

[Suggestion]: Since the title refers to current drug application strategies, current drug application strategies are suggested to be added after listing the existing drugs targeting glucose metabolism enzymes, and then discuss related opportunities and challenges. Arranged by current status,  related opportunities and challenges, and future possibilities as new anticancer drugs are appreciated.   Abstract Page 1, line 21-24 : Sentences are somewhat lengthy, after exploring opportunities and challenges, looking directly at the therapeutic strategies of glucose metabolism modulator in combination with other clinical anticancer drugs may be considered.

Introduction

Page 2: line 39-40: In eukaryotic cells, the yield of ATP produced per glucose is 36 to 38, relying on how the 2 NADH enter the mitochondria. Taken together there are 38 molecules of ATP produced and 2 ATP are formed outside the mitochondria.

Page 3: line 49-51: More references are required to support the strategy of targeting the Warburg effect in cancer cells. Starving cancer cells to death by deprivation of glucose could be considered in the discussion.

Page 4: line 99-100: Please do a thorough literature research since there are existed papers summarizing glucose metabolism enzymes targeted medicinal chemicals (PMID: 35001793).

Drugs targeting glucose transferase (GLUT)

Page 6-7: line 193-201: in this part you have listed varieties of medicinal chemicals as well as natural products inhibiting GLUT. Please compare the synthetic chemicals and natural substances in terms of safety and toxicological issues.

Drugs targeting hexokinase (HK)

Page 7: line 217-219: apart from p53 and p63, p73 also belongs to the same family of related transcription factors, yet they appear to have shared and distinct functions.

        Similarities and differences of how HK-2 is regulated by p53 and p63 should be indicated more specifically.

Page 7: line 228-230: please list supporting literatures.

Drugs targeting phosphofructokinase (PFK)

Page 11: line 370-372: in order to optimise the drug structure for better targeting PKM2, the authors have mentioned nano-formulations. Yet is it your own opinion or it has been reported by solid data?Which type of nano-formulation are suitable for optimization? Dendrimers? polymeric nanoparticles? or nano-emulsions and micelles?

Combinational strategies using glucose metabolism enzyme inhibitors

Page 16: line 579-582: it is highly suggested that you cite the original research article even though compelling results could be obtained from the review paper (reference 263). In case anyone has to solve the problem of elevating redox reactions then he/she has to find the review first then to the original article.

Page 17: line 632-634: As the authors had mentioned in Page 7, line 229-230, the similarities of the structures of different isoforms hinders the small-molecular inhibitors. Different isoforms within a protein family often show remarkable similarity in spatial quaternary structure. Thus the authors are suggested to focus on the enzyme’s binding site of the small molecules in terms of higher selectivity of targeted drugs.

Figure

Figure 1: The differences of reprogramming of glucose metabolism between normal cells and cancer cells can not be clearly distinguished. Merging of the two diverse metabolism or utilising distinct colours and lines to meke them more recognisable may be needed. Streamline and downsize of your figure is also suggested.

Author Response

Dear reviewer,

On behalf of all the contributing authors, I would like to express our sincere appreciation for your constructive comments concerning our manuscript entitled "Targeting glucose metabolism enzymes in cancer treatment: current and emerging strategies" (Manuscript ID: cancers-1904543). These comments are all valuable and helpful in improving our review. Based on your suggestions, we have made some modifications to our manuscript. All corrections are highlighted in blue in the revised manuscript. Please kindly find attached the Revised Manuscript and Point-By-Point Response.

Sincerely yours,

Bingwen Zou M.D.

Sep 14th, 2022
